# Characterization of a Drought-Induced Betaine Aldehyde Dehydrogenase Gene *SgBADH* from *Suaeda glauca*

**DOI:** 10.3390/plants13192716

**Published:** 2024-09-28

**Authors:** Hangxia Jin, Min Tang, Longmin Zhu, Xiaomin Yu, Qinghua Yang, Xujun Fu

**Affiliations:** 1Key Laboratory of Digital Upland Crops of Zhejiang Province, Institute of Crops and Nuclear Technology Utilization, Zhejiang Academy of Agricultural Science, Hangzhou 310021, China; jinhx@mail.ac.cn (H.J.); zlmsllzly@163.com (L.Z.); yuxm@mail.ac.cn (X.Y.); tsingyang2009@163.com (Q.Y.); 2Hangzhou Institute for Food and Drug Control, Hangzhou 310022, China; sugartm1700@163.com

**Keywords:** betaine aldehyde dehydrogenases, *Suaeda glauca*, *Arabidopsis thaliana*, drought stress

## Abstract

Betaine aldehyde dehydrogenases (BADHs) are key enzymes in the biosynthesis of glycine betaine, which is an important organic osmolyte that maintains cell structure and improves plant tolerance to abiotic stresses, especially in halotolerant plants. Improving the drought tolerance of crops will greatly increase their yield. In this study, a novel *BADH* gene named *SgBADH* from *Suaeda glauca* was induced by drought stress or abscisic acid. To explore the biological function of *SgBADH*, the *SgBADH* gene was transformed into Arabidopsis. Then, we found *SgBADH*-overexpressing Arabidopsis seedlings showed enhanced tolerance to drought stress. *SgBADH* transgenic Arabidopsis seedlings also had longer roots compared with controls under drought stress, while *SgBADH*-overexpressing Arabidopsis exhibited increased glycine betaine accumulation and decreased malondialdehyde (MDA) under drought stress. Our results suggest that SgBADH might be a positive regulator in plants during the response to drought.

## 1. Introduction

Plant growth and productivity are strongly influenced by an array of abiotic stresses including drought and salinity. The drought resistance and salinity of crops greatly affects the yield. Glycine betaine (GB), which is one kind of important organic osmolyte, can maintain cell structure and improve plant tolerance to abiotic stresses [1,2,3]. There are also a few non-GB-accumulator plants which do not accumulate glycine betaine, such as rice and Arabidopsis. GB is synthesized through the two-step oxidation of choline catalyzed by choline monooxygenase (CMO) and betaine aldehyde dehydrogenase (BADH) [4,5,6]. As an important enzyme in the second step, BADH catalyzes the oxygenation of betaine aldehyde to GB under stress conditions. Different plant BADHs may exert their effects through different pathways, such as not only acting on this pathway, and BADH catalyzes the production of dimethylsulfoniopropionate by the oxidation of 3-dimethylsulfoniopropionaldehyde [7,8]. These pathways indicate that BADH proteins have conservation, diversity, and complexity in different plants.

BADH belongs to subfamily 10 of the aldehyde dehydrogenase superfamily (ALDH10) [7,8]. Since the BADH gene was first isolated and cloned from *Spinacia oleracea* in higher plants [9], a series of *BADH* genes have been identified from barley [10], spinach [11], *Avena sativa* [12], oilseed rape [13], rice [14,15], Arabidopsis [16], *Halocnermum strobilaceum* [17], and hazelnut [18]. Evidence in the literature demonstrates that BADH in a number of plant species is involved in abiotic stresses, such as spinach, *Avena sativa*, oilseed rape, barley, and sorghum [11,12,13,19,20]. Overexpression of the *BADH* gene is an important strategy to genetically engineer crops to produce GB and increase abiotic stress tolerance. A *Spinacia oleracea BADH* gene (*SoBADH*) was introduced into sweet potato, and then the transgenic sweet potato improved its tolerance to abiotic stresses including salt, oxidative stress, and low temperature [21]. Interestingly, the loss of function of BADH in non-GB-accumulator plants, Arabidopsis and rice, reduced salt stress tolerance or drought stress in transgenic plants [22,23]. These indicate that BADH not only regulates plant stress tolerance through GB but also through other pathways.

*Suaeda glauca*, a succulent halophyte of the Amaranthaceae family, is widely distributed in coastal areas of China [24,25]. *Suaeda glauca* exhibits a high resistance to salt and drought stress and it may provide a valuable source to validate salinity and drought tolerance-related genes to increase crop yield [26]. Two plants of the *Suaeda* genus have been cloned with *BADH* genes [27,28], which respond to stress such as drought and significantly improve plant salt or drought tolerance, fully demonstrating the high potential of *Suaeda* genes to enhance crop growth. However, few genes have been isolated from *Suaeda glauca*, and few reports have explored the molecular regulation in response to abiotic stress in *Suaeda glauca*. Here, a new member of the *BADH* gene family, named *SgBADH* (GenBank: PQ356478), was our first aldehyde dehydrogenase gene identified in *Suaeda glauca*. The BADH protein sequences of *Suaeda salsa* (ABG23669.1), *Suaeda maritime* (AFW04226.1), and *Suaeda liaotungensis* (AAL33906.1) are fully 100% identified, but the identity between the BADH protein of *Suaeda glauca* in this study and *Suaeda liaotungensis* is only 79.4%. Thus, *SgBADH* is a new *BADH* gene in the *Suaeda* genus. In this study, to explore the biological function of *SgBADH*, the sequence analyses, evolutionary analysis, and expression pattern of the *SgBADH* gene were investigated in *Suaeda glauca*. The *SgBADH* gene was transformed into Arabidopsis, and the effect of *SgBADH* overexpression in the transgenic Arabidopsis plants when exposed to drought stress was significant.

## 2. Results

### 2.1. Evolutionary Analysis of SgBADH

We cloned *BADH* cDNAs by PCR using degenerate primers, which were designed on the basis of the highly conserved amino acid sequences from several plants’ *BADH* as described in the Methods section. The nucleotide sequence was named *SgBADH*. The *SgBADH* cDNA contains an open reading frame of 1503 bp encoding 500 amino acids (https://www.ncbi.nlm.nih.gov/nuccore/PQ356478, accessed on 10 September 2024).

The evolutionary relationships of the BADH proteins were analyzed by the multiple alignments of the amino acid sequences from some viruses, archaea, fungi, bacteria, animals, eudicots, and monocots using the neighbor-joining method (Figure 1). The phylogenetic analysis clustered the BADH proteins of the selected species into five major groups. Species from the same realm or class were roughly classified together (same color), but there were also some species belonging to different realms or classes classified together. All BADH proteins of eudicots were located in the same group, while those of monocots were clustered into another group (Figure 1). According to the phylogenetic analysis, the BADH of *Amaranthus tricolor*, *Amaranthus hypochondriacus*, and SgBADH were more similar to each other than to the other BADH proteins of the same *Suaeda* species.

### 2.2. Analysis of the SgBADH Protein Interaction Network Collaborator

The utilization of protein–protein interaction networks to link unknown functional proteins aids in understanding the regulatory networks between biomolecules and predicting the function of unknown proteins. In this study, potential interactions among the SgBADH proteins were explored using the STRING program based on the *Spinacia oleracea* association model (Figure 2). The potential proteins were annotated by Gene Ontology (GO) classification analysis. GO enrichment analysis showed that these interaction network collaborators were mainly enriched in the organonitrogen compound metabolic process, organophosphate metabolic process, lipid metabolic process, glycerophospholipid biosynthetic process, phosphatidylethanolamine biosynthetic process, glycine betaine biosynthetic process from choline, arachidonic acid secretion, and proline metabolic process.

### 2.3. SgBADH Gene Is Stress-Responsive in Suaeda glauca

Mannitol is often used as an osmotic regulator, and its properties can create different degrees of drought for plants. Based on this, this experiment simulated drought stress using a concentration of 200 mM mannitol [29,30]. ABA is an important hormone in plant growth and development, as well as a plant stress hormone accumulated under osmotic imbalance caused by abiotic stress. When plants are subjected to drought stress, they synthesize a large amount of ABA, which leads to the expression of downstream related resistance genes, thereby improving plant drought resistance [31,32]. To examine whether *SgBADH* was induced by abiotic stress in *Suaeda glauca*, *Suaeda glauca* seedlings were exposed to mannitol and abscisic acid (ABA). The gene expression was monitored by qRT-PCR. *SgBADH* was induced by drought stress after 3 days in *Suaeda glauca*, then the transcript level of *SgBADH* decreased after 6 days of drought treatment (Figure 3A). Because *SgBADH* was induced by drought stress, it was of interest to examine whether *SgBADH* was an ABA-dependent stress-responsive gene. The results show that ABA treatment significantly induced the expression of *SgBADH* (Figure 3A). These results indicate up-regulation of *SgBADH* by drought stress might be dependent on ABA in *Suaeda glauca*. Consistent with this trend, the GB content showed a peak after 3 days of drought or ABA treatment and slightly decreased after 6 days.

### 2.4. Expression of SgBADH in Transgenic Arabidopsis

*SgBADH* was overexpressed in Arabidopsis under the control of the *Cauliflower mosaic virus* (CaMV) 35S promoter. The overexpressing *SgBADH* Arabidopsis plant strains were named B1, B2, and B3. Then, the control named CK was an Arabidopsis plant transfected with an empty plasmid. To examine the expression of the *SgBADH* gene, the total RNA isolated from transgenic Arabidopsis leaves was analyzed by qRT-PCR using gene-specific primers. High levels of *SgBADH* transcripts were detected in *SgBADH* transformants, whereas no *SgBADH* transcripts were detected in the control (CK) (Figure 3C). The EGFP transcripts were detected in all of the Arabidopsis transformants, which were transfected with an empty plasmid (CK) or a plasmid containing the *SgBADH* gene (B1, B2, and B3). Based on these data, three lines (B1, B2, and B3) were selected for further analyses.

### 2.5. Overexpression of SgBADH Enhances Drought Tolerance in Transgenic Arabidopsis Seedlings

Seven-day-old seedlings of B1, B2, B3, and the CK Arabidopsis lines were subjected to drought stress (MS + 200 mM D-mannitol) and a normal condition (MS + 0 mM D-mannitol). After two weeks on MS plates with 0 mM D-mannitol, the seedling growth of B1, B2, and B3 was similar with that of the CK seedlings. However, on MS plates with 200 mM D-mannitol, B1, B2, and B3 seedlings grew better than the CK seedlings (Figure 4A). *SgBADH* transgenic Arabidopsis seedlings (B1, B2, and B3) had longer roots compared with CK under drought stress (Figure 4B). These results indicated *SgBADH*-overexpressing Arabidopsis seedlings showed enhanced tolerance to drought stress.

### 2.6. SgBADH-Overexpressing Arabidopsis Exhibits Increased GB Accumulation and Decreased MDA under Drought Stress

*SgBADH*-overexpressing Arabidopsis lines subjected to drought stress (200 mM D-mannitol) for two weeks showed higher GB contents in the leaves and decreased MDA compared with CK plants (Figure 5). This result indicated *SgBADH*-overexpressing Arabidopsis plants show enhanced tolerance to drought stress because of the increase in GB contents.

## 3. Discussion

The phylogenetic analysis of BADH proteins showed that the species from the same realm or class were roughly classified together (same color, Figure 1), but there were also some species belonging to different realms or classes classified together. This means that BADH has relative conservation but also diversity and complexity, which is consistent with other research results [21,33,34,35]. The BADH of *Amaranthus tricolor*, *Amaranthus hypochondriacus*, and SgBADH were more similar to each other than (Figure 1) to the other BADH proteins of the same *Suaeda* species (*Suaeda salsa*, *Suaeda maritime*, *Suaeda liaotungensis*). The BADH protein sequences of *Suaeda salsa* (ABG23669.1), *Suaeda maritime* (AFW04226.1), and *Suaeda liaotungensis* (AAL33906.1) are fully 100% identified, so they cluster together in Figure 1. The similarity in identity of the BADH protein in *Suaeda glauca* in this study and *Suaeda liaotungensis* (AAL33906.1) is 79.4%, while the similarity in identity of the BADH protein in *Suaeda glauca* (SgBADH) and *Amaranthus tricolor* (AUC64503.1) is 89.2%. Though overexpression of *SlBADH* (from *Suaeda liaotungensis*) improved tolerance to salinity in transgenic tobacco, this result suggests that SgBADH might have a different mechanism of GB production than other *Suaeda* plants.

The predicted SgBADH protein interaction collaborators (Figure 2) were mainly enriched in the glycerophospholipid biosynthetic process, phosphatidylethanolamine biosynthetic process, glycine betaine biosynthetic process from choline, arachidonic acid secretion, and proline metabolic process, which were related to drought stress [36,37,38,39,40,41,42,43]. This is consistent with *SgBADH* expression patterns under drought stress and the drought tolerance of *SgBADH*-overexpressing Arabidopsis plants.

In previous studies on the expression characteristics of the *BADH* gene, it was found that the *BADH* gene in different plants was either constitutively expressed or induced, which may be related to the specific function of the gene in different plants. In plants such as rice, spinach, and barley, two *BADH* genes were found, some of which have different expression patterns [2,5,44]. For example, in response to salt treatments of increasing concentrations, the level of the *OsBADH1* (LOC4336081) transcript increased significantly in rice, while no consistent relationship between *OsBADH2* (coding BAC76608.1 in Figure 1) transcript levels and salt treatment was observed, suggesting that *OsBADH1* but not *OsBADH2* has a role in the response of rice to salt stress [44]. The two *BADH* genes *BBD1* and *BBD2* in barley were induced by salt, drought, and ABA treatment with different levels [45,46]. This may imply that the two *BADH* homologous genes in the same plant have different roles in the abiotic stress response. Limited by the absence of a reference genome for *Suaeda glauca*, we used degenerate primers designed based on homologous sequences to clone only one *BADH* gene from *Suaeda glauca*. Compared with *OsBADH2* (percent identity: 72.3%), *SgBADH* has a higher similarity to *OsBADH1* (percent identity: 80.2%) in terms of nucleotide sequence alignment, and we found that the expression pattern of *SgBADH* is similar to that of one of the *BADH* genes in the aforementioned plants (*OsBADH1*), which means that *SgBADH* might be the *BADH* gene that responds to adversity, if there are two *BADH* genes in *Suaeda glauca.*

As an organic osmolyte, GB has been shown to increase the tolerance to abiotic stresses in transgenic plants [1,21,47]. In this study, the overexpression of *SgBADH* in Arabidopsis increases *BADH* expression and GB accumulation in vivo and, consequently, leads to an enhanced tolerance of the transgenic Arabidopsis to drought stress (Figure 4 and Figure 4). Under drought stress, the *SgBADH* transgenic Arabidopsis lines (B1–B3) accumulated approximately 2–3 times more GB content than that of the control (Figure 5A). The MDA content was reduced in *SgBADH* transgenic Arabidopsis lines (B1–B3) under drought stress (Figure 5B), which suggests that cell membrane homeostasis was improved in the *SgBADH* transgenic Arabidopsis lines. Therefore, this indicates that GB protects the cell membrane from stress-induced injury and maintains the osmotic balance [1,2,48].

The present study suggests that the expression of the *SgBADH* gene and the accumulation of glycine betaine might be two of the molecular mechanisms that indicate how SgBADH copes with drought stress. This study indicates the potential of the *SgBADH* gene for application to the genetic engineering of crops for the improvement in drought tolerance.

## 4. Materials and Methods

### 4.1. Plant Materials and Salt Stress Treatment

*Suaeda glauca* plants were grown in a greenhouse under a 16:8 h light–dark cycle at Zhejiang Academy of Agricultural Science, Hangzhou, China. One-month-old seedlings were treated with 200 mM mannitol or 100 µM ABA for three days or six days. The shoots of the treated seedlings and three untreated seedlings were immediately frozen and stored in liquid nitrogen for RNA extraction.

Seven-day-old seedlings of transgenic Arabidopsis lines were subjected to drought stress (MS + 200 mM D-mannitol) and a normal condition (MS + 0 mM D-mannitol) for two weeks.

### 4.2. RNA Extraction, cDNA Synthesis, and Cloning of SgBADH cDNAs

The total RNA was extracted from the roots of plant samples. First-strand cDNA was synthesized using oligo (dT) primers (TOYOBO, Osaka, Japan). The fragment of *SgBADH* cDNA was amplified by PCR from cDNA of *Suaeda glauca* using specific primers designed on the basis of the highly conserved amino acid sequences from several plant BADH cDNAs of AAB41696.1 (*Spinacia oleracea*), AAG51938.1 (*Arabidopsis thaliana*), and a cDNA library of the leaves of salt-treated *Suaeda glauca*. The primers used in this study are listed in Table 1 (SgBADHF/SgBADHR). Then, cDNA was sequenced in Beijing Tsingke Biotech Co., Ltd., Beijing, China.

### 4.3. qRT-PCR Analysis

The *SgBADH* gene and the internal reference *actin* gene were subjected to real-time quantitative PCR (qRT-PCR) with specific primers, which were designed in Primer-BLAST (https://www.ncbi.nlm.nih.gov/tools/primer-blast/, accessed on 22 September 2016). qRT-PCR analysis of the two genes was conducted in triplicate with the SYBR Green Real-time PCR Master Mix (TOYOBO, Osaka, Japan) on the LightCycler^®^ 480 Real-Time PCR System (Roche Diagnostics GmbH, Mannheim, Germany). The primers used in this study are listed in Table 1.

### 4.4. Phylogenetic Tree and Protein–Protein Interaction Analysis

A phylogenetic analysis based on 93 full-length amino acid sequences (search in https://www.ncbi.nlm.nih.gov/protein/, accessed on 5 June 2024) including SgBADH was run with MEGA7.0. The analysis was subjected to the neighbor-joining method, and the bootstrap method supported by 1000 replicates to test phylogeny. The functional relationships of the SgBADH protein were predicted using the STRING protein interaction database (https://string-db.org/, accessed on 5 June 2024) (STRING11.5). The STRING database integrates protein interactions from experimental data, the literature, and predictions. Gene Ontology (GO) classification analysis was performed based on the protein analysis by STRING11.5.

### 4.5. Plasmid Construction, Arabidopsis Transformation

The coding region of *SgBADH* was obtained using RT-PCR from *SgBADH* with specific primers *SgBADH*-F/*SgBADH*-R (Table 1). The PCR product was cloned into a linearized pCAM35S::EGFP expression vector (containing the HTP gene for selection and the EGFP gene) using the In-Fusion HD Kit (Clontech, Mountain View, CA, USA). The recombinant plasmid vector was named pCAM35S::*SgBADH*::EGFP. The pCAM35S::*SgBADH*:: EGFP and pCAM35S::EGFP (empty plasmid as the control) vector were introduced into *Agrobacterium tumefaciens* strain EHA105 via the freeze–thaw method, respectively. The expression vectors were then transformed into Arabidopsis (*Columbia*) using the floral dip method [49].

### 4.6. Verification of SgBADH-Overexpressing Transgenic Arabidopsis

*SgBADH*-overexpressing transgenic lines were selected on MS with 50 mg/L Hygromycin B. T2 generation plants of the transgenic lines B1, B2, and B3 were used in subsequent experiments. The Arabidopsis plants were grown in a climate chamber with 16:8 h light–dark cycle at 23 °C. RNAs isolated from transgenic Arabidopsis lines were used as templates for qRT-PCR and amplified using specific primers of the *SgBADH* gene: *SgBADH*192-F/*SgBADH*192-R. (Table 1), which was designed in Primer-BLAST (https://www.ncbi.nlm.nih.gov/tools/primer-blast/, accessed on 22 September 2016). Another pair of specific primers EGFP-F/EGFP-R (Table 1) were used to amplify the 145 bp fragment of an *EGFP* gene.

### 4.7. Analysis of the Drought Tolerance in SgBADH-Overexpressing Transgenic Arabidopsis

Seeds of T2 transgenic Arabidopsis lines and the control (transgenic empty plasmid Arabidopsis line) were surface-sterilized and then sown on solid MS medium with 50 mg/L Hygromycin B. After 7 days, the seedlings with equivalent growth vigor were selected and transplanted onto solid MS medium supplemented with 0 and 200 mM D-mannitol. After 2 weeks, the root growth was measured. The root lengths were determined in 20 plants of each line of transgenic and control Arabidopsis. The plants grew at 25 °C under a 16 h light/8 h dark photoperiod.

### 4.8. Lipid Peroxidation Assay

The levels of lipid peroxidation were determined by the amounts of malondialdehyde (MDA) as the end product of lipid peroxidation in plant cells. The MDA content was measured using the thiobarbituric acid method [50,51,52]. Thus, 1 g of fresh tissue was homogenized in 10 mL of 10% (*W*/*V*) trichloroacetic acid and centrifuged at 4000× *g*. The supernatant was mixed with an equal volume of 2-thiobarbituric acid (0.67% in 10% [*w*/*v*] trichloroacetic acid) and then the mixture was boiled for 15 min and centrifuged to clarify the solution. Absorbance of the supernatant was measured at 532 nm and corrected for interference by subtracting the A600 and A450. The MDA content was calculated using the formula: MDA (μmol/L)= 6.45 × (A532 − A600) − 0.56 × A450.

### 4.9. Measurement of GB Content

The GB contents were measured according to the protocols described by Martinez [53] with some modifications. Fresh leaves (1 g) of the plants were ground in water and incubated for 5 h at 70 °C and then centrifuged for 10 min at 4000× *g*. The supernatants were divided into two portions; one portion was adjusted to a pH 1.0 to determine the amount of GB contents and choline contents (V3), and the other was adjusted to a pH 8.0 to determine the choline contents (V2). Under acidic conditions, choline and betaine reineckate are precipitated, and at an alkaline pH, only choline reineckate precipitates. The supernatants were mixed with 5 mL 3% Reinecke’s salt (pH 1.0 or pH 8.0), centrifuged, and the supernatants were discarded. Then, the residues were washed with ether and dissolved in 70% acetone. The absorbance was measured at 525 nm (HITACHI J-2800, Hitachi, Tokyo, Japan), and V3 and V2 were determined using a standard curve, respectively. Three replicate biological experiments were conducted. GB content (V1) = Amount (V3) − choline content (V2).

## Figures and Tables

**Figure 1 plants-13-02716-f001:**
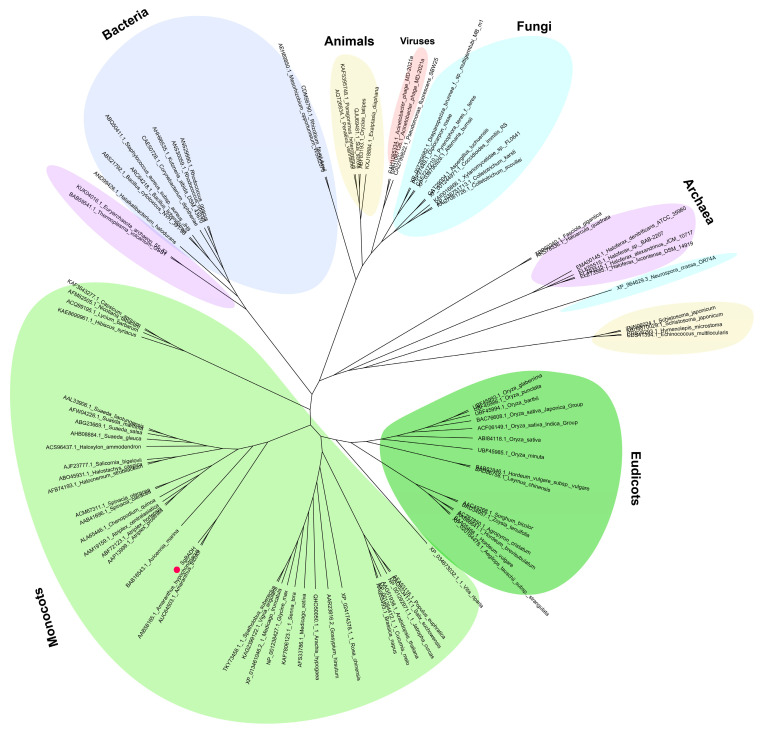
Phylogenetic tree of BADH proteins. The neighbor-joining (NJ) phylogenetic tree was constructed using the MEGA7 with 1000 bootstrap replicates. The red symbol is SgBADH protein.

**Figure 2 plants-13-02716-f002:**
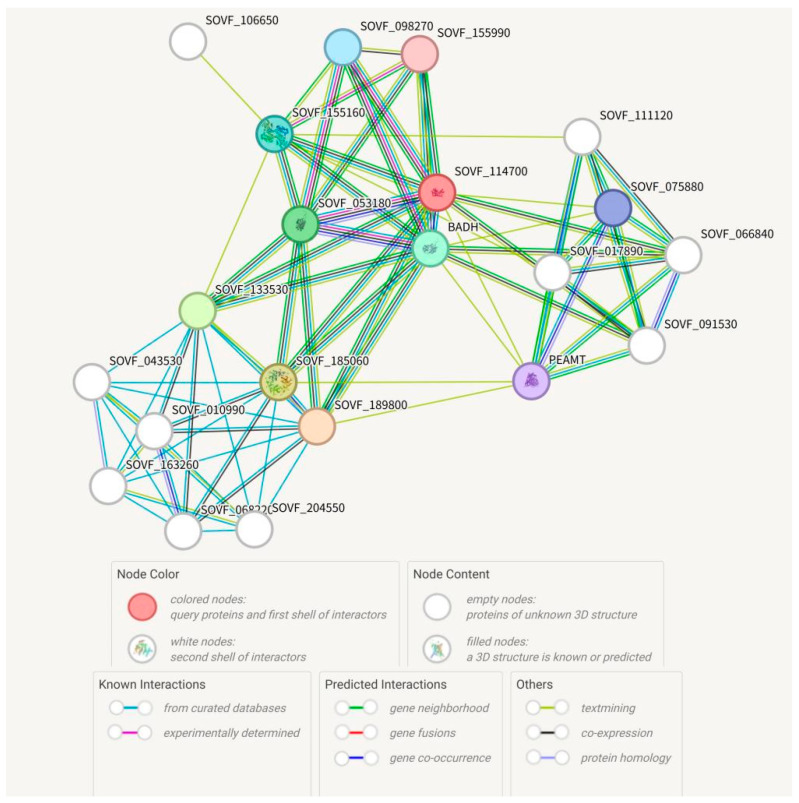
Protein–protein interaction network of SgBADH protein based on its ortholog in *Spinacia oleracea*. Red node (SOVF_114700) is the ortholog of SgBADH. Nodes represent proteins. Connections between nodes represent interactions between proteins, with edge thickness indicating the confidence level of the interaction.

**Figure 3 plants-13-02716-f003:**
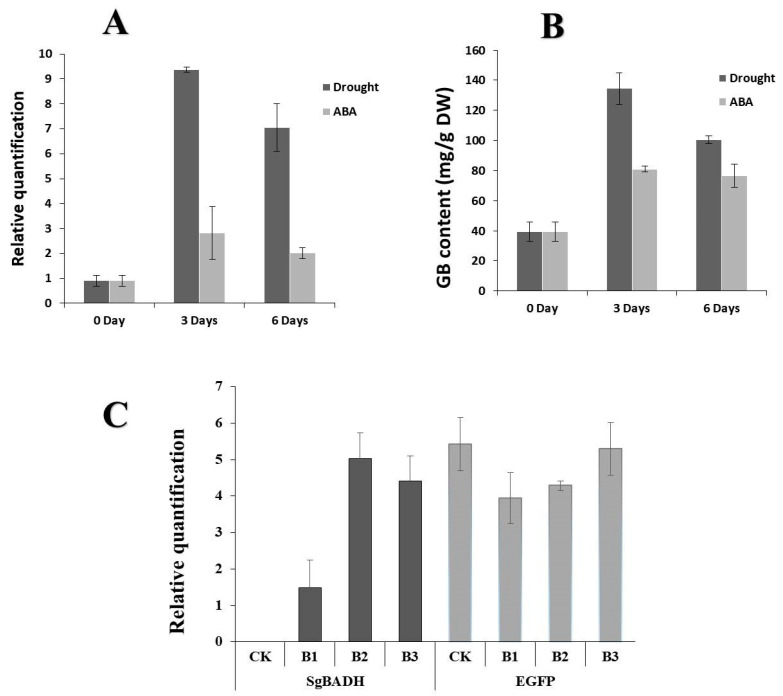
qRT-PCR analysis and GB levels of *Suaeda glauca*. (**A**) *SgBADH* induction by drought and ABA treatments in *Suaeda glauca*. *Suaeda glauca* seedlings were treated with 200 mM mannitol or 100 µM abscisic acid (ABA) for three days and six days. (**B**) GB levels were measured during treatment with 200 mM mannitol or 100 µM ABA treatment for three days and six days in *Suaeda glauca*. (**C**) *SgBADH* and *EGFP* expression levels in transgenic Arabidopsis (B1–B3) and CK which was transfected with an empty plasmid. Three replicates of each reaction were performed. The expression values were calculated using the 2^−ΔΔCt^ method and the actin gene as an endogenous control. The bars represent the standard error of the mean.

**Figure 4 plants-13-02716-f004:**
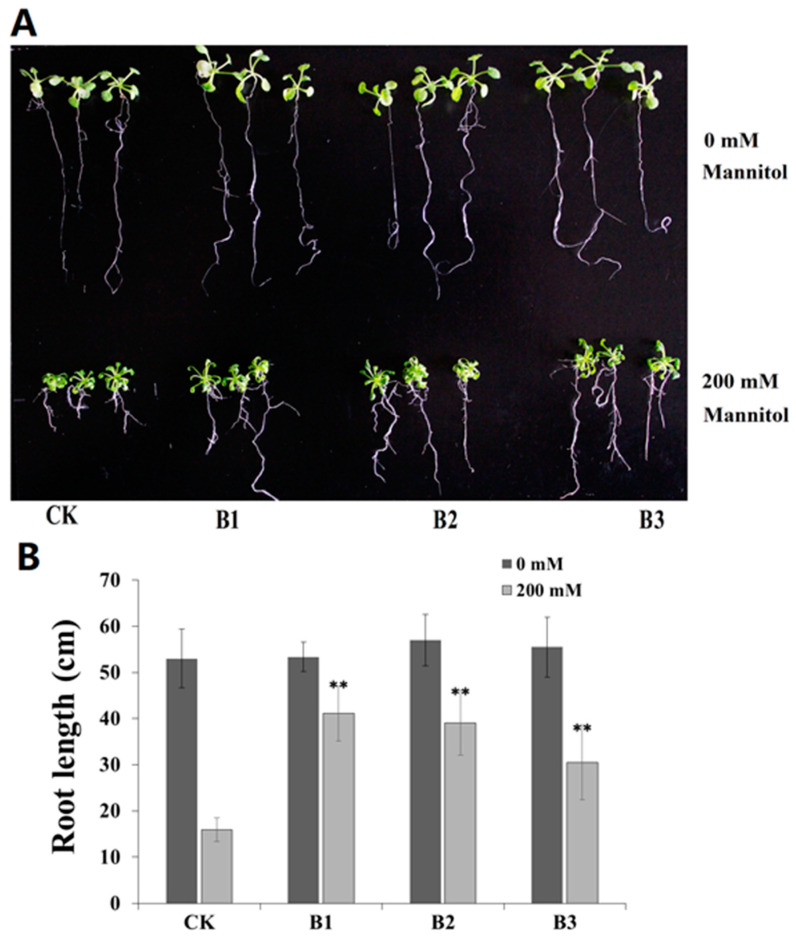
*SgBADH* transgenic and control Arabidopsis grown on MS media containing 200 mM mannitol. Morphology (**A**) and root length (**B**) of *SgBADH*-overexpressing Arabidopsis seedlings grown for two weeks on MS medium containing 0 mM and 200 mM D-mannitol. ** indicates significant differences from the control at *p* < 0.01.

**Figure 5 plants-13-02716-f005:**
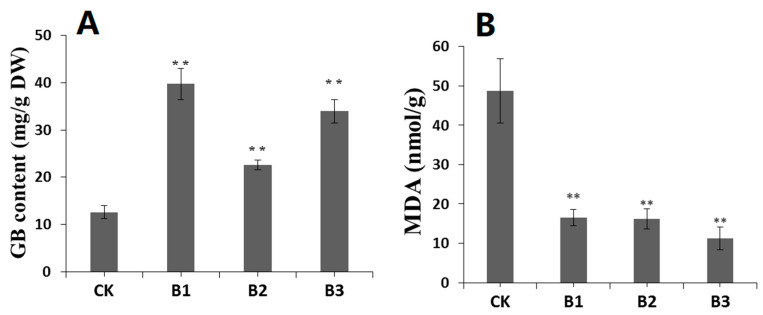
GB (**A**) and MDA (**B**) content in Arabidopsis plants under drought stress. GB and MDA were extracted from the tissues of the 14-day drought treatment. ** indicates significant differences from the control at *p* < 0.01.

**Table 1 plants-13-02716-t001:** Sequences of primers used in this study.

Gene	Primer Sequences
*SgBADHF*	ATGKCGWTCCCWATWCCTKCTCG
*SgBADHR*	TYAAGGAGACTTGTACCAYCCC
*SgBADH-F*	ATGGCGTTCCCAATTCCTGC
*SgBADH-R*	TCAAGGCGACTTGTACCATC
*SgBADH192-F*	AAATATGGAAGGAGGAAGT
*SgBADH192-R*	GTTGTGAGCAATTAACCC
*AtActin124-F*	ATCGCTGACCGTATGAG
*AtActin124-R*	TGAGGGAAGCAAGAATG
*SgActin-F*	CCGCAAAGATTACATACC
*SgActin-R*	TCACCGAAAGTGCTTCTA
*EGFP-F*	ACCCTCGTGACCACCCTGAC
*EGFP-R*	TGTAGTTGCCGTCGTCCTTGA

## Data Availability

Date are contained within the article.

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
