# Peer review of "Characterization of a Drought-Induced Betaine Aldehyde Dehydrogenase Gene SgBADH from Suaeda glauca"

_plants, 2024, doi:10.3390/plants13192716_

Round 1

Reviewer 1 Report

Comments and Suggestions for Authors

I have reviwed the manuscript. First i would like that the authors would clearly state the novelty of the study. 

Figure 1 should be provided in better resoultion.

The refreneces do not follow the MDPI style.

Author Response

Comments 1: First i would like that the authors would clearly state the novelty of the study.

Response 1: Thanks for your comments on our manuscript. We have revised the paper according to the comment in the Introduction.

Comments 2:   Figure 1 should be provided in better resoultion.

Response 2: Thanks for your comments on our manuscript. We have revised the figure 1 according to the comment.

Comments 3:  The references do not follow the MDPI style.

Response 3: Thanks for your comments on our manuscript. We have revised the references style.

Reviewer 2 Report

Comments and Suggestions for Authors

I found this article interesting but I have few important suggestion.

The introduction should be extended – it is not clear whether Arabidopsis thaliana possesses similar to  BADH genes and what is their role.  I found in the discussion that some crops possess (  rice, spinach, and barley)  BADH  homologs but there is no information whether their function in drought stress have been investigated.     

The description of the results must be  deeply clarified. It is not clear what kind of seedlings have of Suaeda glauca been used The name suggests that it is a mutant BADH lines but as far as I understood it is probably wild type plant.

I found the term Soybean seedlings in Figure 3 but the gene derived from the Suaeda glauca so I am not sure what kind of plants were used.

The description of the Figure 3 panel B is also unclear.  I found at the next page what CK B1, B2, B3 variants mean.

Moreover, I know what is the role of the mannitol but its  role was not clearly explained at the beginning for potential readers The same situation is with ABA.

The authors shows the mRNA expression but not always high transcription level leads to high protein content  - it should be verified via western blotting and/or deeply discussed. Authors used plasmid with GFP so why there is no experiments showing their fluorescence in cells – such experiments would simply confirmed efficient translation and localization.

I think that manuscript would be also improved by the analysis of the proline content which is usually used as drought stress indicator.  The authors checked the level of  mRNA BADH after 3 days, but it was be interesting to check how long increased expression is maintained,  for example what happened after 14 days with this expression and the level of GB, MDA and proline. In my opinion such prolonged experiments would answer the important the question about stability of the high expression – it is crucial  for  commercial use what was suggested as the goal of this study.   

Author Response

Comments 1: The introduction should be extended – it is not clear whether Arabidopsis thaliana possesses similar to BADH genes and what is their role. I found in the discussion that some crops possess (rice, spinach, and barley) BADH homologs but there is no information whether their function in drought stress have been investigated.  

Response 1: Thanks for your comments on our manuscript. We have revised the paper according to the comment in Introduction.   

Comments 2: The description of the results must be deeply clarified. It is not clear what kind of seedlings have of Suaeda glauca been used. The name suggests that it is a mutant BADH lines but as far as I understood it is probably wild type plant.

Response 2: Thanks for your comments on our manuscript. We have revised the paper according to the comment in figure 3 legend and section ‘2.4. Expression of SgBADH in transgenic Arabidopsis’.

Comments 3: I found the term Soybean seedlings in Figure 3 but the gene derived from the Suaeda glauca so I am not sure what kind of plants were used.

Response 3: Thanks for your comments on our manuscript. We wrote it wrongly. We have revised the word ‘Soybean’ to Suaeda glauca.

Comments 4: The description of the Figure 3 panel B is also unclear.  I found at the next page what CK B1, B2, B3 variants mean.

Response 4: Thanks for your comments on our manuscript. We have revised the paper in figure 3 legend and section ‘2.4. Expression of SgBADH in transgenic Arabidopsis’.

Comments 5: I know what is the role of the mannitol but its role was not clearly explained at the beginning for potential readers The same situation is with ABA.

Response 5: Thanks for your comments on our manuscript. We have revised the paper according to the comment in section ‘2.3. SgBADH gene is stress-responsive in Suaeda glauca’.

Comments 6: The authors shows the mRNA expression but not always high transcription level leads to high protein content - it should be verified via western blotting and/or deeply discussed. Authors used plasmid with GFP so why there is no experiments showing their fluorescence in cells – such experiments would simply confirmed efficient translation and localization.

Response 6:

Thanks for your comments on our manuscript. Although we did not measure the content of BADH protein, the expression pattern of BADH gene and the trend of changes in the product GB (Added in Figure 3) catalyzed by BADH enzyme are consistent, which demonstrates the gene fuction.

In fact, we conducted this experiment and the results are shown in the following figure. The results seem to be localized to the cell membrane or peroxisome, but we did not conduct a control for peroxisome localization. Because this experiment was conducted around 2017, there is currently a shortage of raw materials that cannot be replenished. When we are currently working on cloning another BADH, we may compare and discuss this issue in another article in the future.

Comments 7: I think that manuscript would be also improved by the analysis of the proline content which is usually used as drought stress indicator. The authors checked the level of  mRNA BADH after 3 days, but it was be interesting to check how long increased expression is maintained,  for example what happened after 14 days with this expression and the level of GB, MDA and proline. In my opinion such prolonged experiments would answer the important the question about stability of the high expression – it is crucial for commercial use what was suggested as the goal of this study.

Response 7: Thanks for your comments on our manuscript. We ever test GB content of Suaeda glauca around 2017, now we added it in Figure 3B. The similar trend of expression pattern of SgBADH gene and the changes in the product GB (Added in Figure 3) catalyzed by BADH enzyme after drought and ABA treatment, combined the phenomenon of SgBADH transgenetic Arabidopsis, demonstrate the gene function in the GB synthesis. We did not test proline content, because SgBADH is the main enzyme in GB synthesis, so we focus on the SgBADH expression and GB content. And there is currently a shortage of raw materials that cannot be replenished.

Reviewer 3 Report

Comments and Suggestions for Authors

This manuscript describes the study of the betaine aldehyde dehydrogenase gene SgBADH from Suaeda glauca, the expression of which can be induced by drought stress and abscisic acid treatment. The authors obtained transgenic Arabidopsis plants overexpressing the SgBADH gene and showed that such plants acquired increased drought tolerance, and under drought conditions formed longer roots and accumulated increased concentrations of glycine betaine (organic osmolyte) and decreased amount of MDA. This may indicate the involvement of SgBADH in the positive regulation of the defense response of plants to drought.

It would be nice to indicate the name of this gene in the title of the manuscript.

The references are relevant to the content, but are provided throughout the text in a format that is incorrect for the journal's requirements. Some of the references given in the text are not included in the bibliography (for example, Sheen et al. 2013, Line 28). Authors need to format links correctly and check their availability.

The Abstract clearly summarizes the content of the research.

In the Introduction, the authors provide a brief description of the Suaeda glauca and the BADH gene (the main function, plant species where these genes have been identified, participation in the stress response). L21: “Different plant BADHs” – “BAHD gene form different plant species” (It's better to write it this way, otherwise the idea arises that there is a whole family of such genes in the plant).

There are at least two papers (found in the NCBI PubMed) characterizing BADH genes from two related Saeda species (see below), including BADH response to stresses. These data should be briefly summarized in the Introduction and discussed in Discussion in comparison with the results obtained in the present paper.

1. Wang FW, Wang ML, Guo C, Wang N, Li XW, Chen H, Dong YY, Chen XF, Wang ZM, Li HY. Cloning and characterization of a novel betaine aldehyde dehydrogenase gene from Suaeda corniculata. Genet Mol Res. 2016 Jun 20;15(2). doi: 10.4238/gmr.15027848.

2. Li QL, Gao XR, Yu XH, Wang XZ, An LJ. Molecular cloning and characterization of betaine aldehyde dehydrogenase gene from Suaeda liaotungensis and its use in improved tolerance to salinity in transgenic tobacco. Biotechnol Lett. 2003 Sep;25(17):1431-6. doi: 10.1023/a:1025003628446.

In the first section (2.1) of the Results, it would be recommended to provide the identification characteristics of the SgBADH gene in the database, and the Internet address of this database. The gene accession number is also missing from the dendrogram in Fig. 1.

The dendrogram includes BADH genes from other Suaeda species. Please indicate which of these genes correspond to the BADH genes described in papers 1 and 2 (see above), so that these genes can then be compared with the gene analyzed by the authors. The comparison should be included in the Discussion chapter.

Section 2.3, L93: “SgBADH seedlings…” Do you mean Suaeda glauca seedlings?

Fig. 3, L101: “Soybean seedlings” – Soybean or Suaeda glauca?

In Discussion, L158-173: In the text, it is advisable to clarify which gene version - BADH1 or BADH2 - is structurally and phylogenetically closer to the gene SgBADH studied by the authors.

The materials and methods are given in sufficient detail and can be reproduced on this basis. In section 4.9, it should be added how exactly the choline content was determined.

Minor:

L81: “proteins was annotated” – “… were…”

L82: “Go enrichment analysis” – “GO enrichment…”

The names of plant species and genes are not always written in italics (see Lines 21, 41, 159-170, 176, 204, 205, 225).

Overall, the presented manuscript briefly but clearly presents the study conducted by the authors on the functional role of the SgBADH gene in determining drought resistance in Suaeda glauca. The results of the work are important for use in crop breeding and can be accepted for publication after minor edits to the text.

Comments on the Quality of English Language

The meaning of the presented material is clear, but editing is needed.

Author Response

Comments 1:  It would be nice to indicate the name of this gene in the title of the manuscript.

Response 1: Thanks for your comments on our manuscript. We have added the name of this gene in the title of the manuscript.

Comments 2:  The references are relevant to the content, but are provided throughout the text in a format that is incorrect for the journal's requirements. Some of the references given in the text are not included in the bibliography (for example, Sheen et al. 2013, Line 28). Authors need to format links correctly and check their availability.

Response 2: Thanks for your comments on our manuscript. We have revised the references.

Comments 3:  The Abstract clearly summarizes the content of the research.

Response 3: Thanks for your comments on our manuscript.

Comments 4: There are at least two papers (found in the NCBI PubMed) characterizing BADH genes from two related Suaeda species (see below), including BADH response to stresses. These data should be briefly summarized in the Introduction and discussed in Discussion in comparison with the results obtained in the present paper.

Response 4: Thanks for your comments on our manuscript. We have added it in Introduction and discussed in Discussion.

Comments 5:  In the first section (2.1) of the Results, it would be recommended to provide the identification characteristics of the SgBADH gene in the database, and the Internet address of this database. The gene accession number is also missing from the dendrogram in Fig. 1.

Response 5: Thanks for your comments on our manuscript. We have added the identification characteristics of the SgBADH gene and the Internet address in the first section 2.1. In order to highlight SgBADH in the image, we specifically labeled it with the gene name in the Figure 1. And we list the accession number in the introduction.

Comments 6:  The dendrogram includes BADH genes from other Suaeda species. Please indicate which of these genes correspond to the BADH genes described in papers 1 and 2 (see above), so that these genes can then be compared with the gene analyzed by the authors. The comparison should be included in the Discussion chapter.

Response 6: Thanks for your comments on our manuscript. We have added the discussion in section Discussion.

Comments 7:  Section 2.3, L93: “SgBADH seedlings…” Do you mean Suaeda glauca seedlings?

Response 7: Yes, it is ‘Suaeda glauca seedlings’, we have modified it in Section 2.3.

Comments 8: Fig. 3, L101: “Soybean seedlings” – Soybean or Suaeda glauca?

Response 8: It is ‘Suaeda glauca seedlings’, we have modified it in Fig. 3 legend.

Comments 9:  In Discussion, L158-173: In the text, it is advisable to clarify which gene version - BADH1 or BADH2 - is structurally and phylogenetically closer to the gene SgBADH studied by the authors.

Response 9: Thanks for your comments on our manuscript. We have added in the discussion.

Comments 10: The materials and methods are given in sufficient detail and can be reproduced on this basis. In section 4.9, it should be added how exactly the choline content was determined.

Response 10: Thanks for your comments on our manuscript. Our description in the method is relatively concise, including the detection method of choline. We have revised some sentences in section 4.9.

Comments 11:  L81: “proteins was annotated” – “… were…”

Response 11: Thanks for your comments on our manuscript. We have revised it.

Comments 12:  L82: “Go enrichment analysis” – “GO enrichment…”

Response 12: Thanks for your comments on our manuscript. We have revised it.

Comments 13:  The names of plant species and genes are not always written in italics (see Lines 21, 41, 159-170, 176, 204, 205, 225).

Response 13: Thanks for your comments on our manuscript. We have revised them.

Reviewer 4 Report

Comments and Suggestions for Authors

The article is reasonably well written, and the experimental strategy is adequate and well executed. However, it requires minor corrections and improvements:

The authors used degenerate primers designed based on homologous sequences of other species [AAB41696.1 (Spinacia oleracea), AAG51938.1 (Arabidopsis thaliana)] to amplify the cDNA of SgBADH using PCR. However, it is not clear how the primers for the rRT-PCR assays were designed. It is important to mention in the text whether the cDNA of SgBADH was sequenced.

It is also important to note that previous mRNA sequences of Suaeda glauca betaine aldehyde dehydrogenase have been reported. For example, HS411192.1 (2010) and KF594413.1 (2013). Are these sequences used in this study?

It is difficult to fully interpret the results of the phylogenetic analysis of BADH proteins because the readability/resolution presented in Figure 1 does not allow the reading of the identifiers of the analyzed sequences. Please improve the resolution of Figure 1.

The text does not describe how and where the sequences for phylogenetic analysis were obtained.

The authors present scientific claims without proper references in the text. For example, the paragraph in lines 158-173 presents several scientific facts from previous studies without citing relevant references. Please refer to previous scientific literature.

Minor Comments:
Line 19. Please state the full name of the substance: malondialdehyde (MDA).
Line 101. Why "Soybean seedlings”?

Comments on the Quality of English Language

Minor editing of English language are required.

Author Response

Comments 1: The authors used degenerate primers designed based on homologous sequences of other species [AAB41696.1 (Spinacia oleracea), AAG51938.1 (Arabidopsis thaliana)] to amplify the cDNA of SgBADH using PCR. However, it is not clear how the primers for the qRT-PCR assays were designed. It is important to mention in the text whether the cDNA of SgBADH was sequenced.
Response 1: Thanks for your comments on our manuscript. We have added the software in section 4.3 and 4.6.

Comments 2:  It is also important to note that previous mRNA sequences of Suaeda glauca betaine aldehyde dehydrogenase have been reported. For example, HS411192.1 (2010) and KF594413.1 (2013). Are these sequences used in this study?
Response 2: HS411192.1 (2010) may be half part of betaine aldehyde dehydrogenase gene, which was not used in this study.  KF594413.1 (2013), which we used in phylogenetic tree of BADH proteins, did not discuss in this study. We cannot confirm whether this sequence belongs to Suaeda glauca or not because its organism name was Suaeda glauca in Genbank, but the link article was ‘Exploration for the salt stress tolerance genes from a salt-treated halophyte, Suaeda asparagoides’.

Comments 3:  It is difficult to fully interpret the results of the phylogenetic analysis of BADH proteins because the readability/resolution presented in Figure 1 does not allow the reading of the identifiers of the analyzed sequences. Please improve the resolution of Figure 1.
Response 3: Thanks for your comments on our manuscript. We have revised the figure 1 according to the comment.

Comments 4:  The text does not describe how and where the sequences for phylogenetic analysis were obtained.
Response 4: Thanks for your comments on our manuscript. We have added it in section 4.4.

Comments 5:  The authors present scientific claims without proper references in the text. For example, the paragraph in lines 158-173 presents several scientific facts from previous studies without citing relevant references. Please refer to previous scientific literature.
Response 5: Thanks for your comments on our manuscript. We have cited the references.

Comments 6:  Line 19. Please state the full name of the substance: malondialdehyde (MDA).

Response 6: Thanks for your comments on our manuscript. We have revised it.

Comments 7:  Line 101. Why "Soybean seedlings”?

Response 7: Thanks for your comments on our manuscript. We wrote it wrongly. We have revised the word ‘Soybean’ to Suaeda glauca.

Round 2

Reviewer 2 Report

Comments and Suggestions for Authors

The manuscript looks much better. I think that it is ready for publication.